



# Technical note: Transit time distributions are not L-shaped

Earl Bardsley[1]

[1]School of Science, University of Waikato, Hamilton 3240, New Zealand

*Correspondence to*: Earl Bardsley (earl.bardsley@waikato.ac.nz)

**Abstract.** A probability density function $f(t)$ with origin at $t = 0$ is defined here as being "L-shaped" if $f'(t) \leq 0$ for $t \geq 0$. L-shaped probability density functions, especially exponential distributions, are often assumed as transit time distributions in hydrological studies. However, L-shaped transit time distributions are not possible. This is because the transit time of a particle must always be with reference to a store, the transit time being some finite duration of time between particle entry and exit. Tracer particles cannot transit through any part of a store in zero time so transit time distributions have the property

$f(0) = 0$, which is incompatible with L-shaped probability density functions. This is a fundamental constraint on the form of transit time distributions, which must possess at least one mode at $t > 0$. Some L-shaped probability density functions may well approximate actual transit time distributions, but they are of different form to the true distributions. A call is therefore made for L-shaped probability density functions to be no longer employed in transit time modelling.

**1 Introduction**

This technical note is concerned with constraints on the form of transit time distributions, with application to hydrology. Attention is drawn to the fact that transit time distributions in nature cannot be L-shaped, where a probability density function $f(t)$ with origin at $t = 0$ is defined here to be L-shaped if $f'(t) \leq 0$ for $t \geq 0$. Thus L-shaped distributions cannot have any modes for $t > 0$. L-shaped transit time distributions which have been assumed in hydrological studies include

exponential distributions and also gamma distributions with shape parameter values < 1.

By way of illustration, we can consider an idealised catchment tracer experiment with a perfect recorder detecting every tracer particle departing within the stream discharge at a point at the lower end of the catchment. Also, every tracer particle which enters the catchment leaves sooner or later via the recorder.


In an instant of time, distribute $N$ tracer particles over the extent of the catchment. In the same instant of time place $M$ tracer particles of the same type exactly onto the recorder itself. The $N$ tracer particles eventually all depart the catchment via stream discharge at varying times $t_1, t_2 \dots t_N$.





On the basis of this idealised experiment, does Eq. (1) or Eq. (2), give the correct expression for catchment mean transit time $\mu$?

$$\mu = \sum_{i=1}^{N} t_i / N \tag{1}$$

$$\mu = \sum_{i=1}^{N} t_i /(N+M) \tag{2}$$

It is of course Eq. (1) which is correct because the $M$ tracer particles placed onto the recorder at $t = 0$ never transited through any part of the catchment system and therefore and have no connection to catchment transit times. The physical analogy would be a drop of rain containing tracer particles falling onto the recorder at the start of the experiment.

Put another way, it is not possible to have transit times of exactly zero because any tracer particles initially present on the recorder have never entered the store concerned. That is, they did not transit to the recorder. It follows that any transit time distribution $f(t)$ in nature must have the property $f(0) = 0$, which has the meaning of zero probability of a transit time in the small interval $0, 0 + dt$.

The above is a purely conceptual argument and not predicated on the feasibility of an actual measurement procedure. In particular, there is no implication of a need to specify some form of physical exclusion zone around an actual recording site in order to define a set of particles "at" the recorder at time zero.

It may seem a conceptual irrelevancy to make the distinction between the unobservable situations of the first tracer particle
arriving fractionally after time zero as opposed to exactly at time zero. However, $f(0) = 0$ has immediate consequence because it follows that all transit time distributions must have at least one mode for $t > 0$. Given this knowledge, it is better science to model transit time distributions by choosing probability density functions which have $f(0) = 0$ and hence possess one or more modes for $t > 0$, consistent with the true situation. A call is made therefore to abandon use of L-shaped probability density functions for transit time modelling. If necessary, finite mixtures of distributions with $f(0) = 0$ could be
employed to give the required degree of fit to data.

An argument in support of L-shaped distributions might be made along pragmatic lines that they have served their purpose through many transit time studies and this longevity justifies their continued use. However, ability to fit data does not in itself equate to theoretical justification and the absence of a mode in L-shaped probability density functions means that there
must have been some degree of error in their initial descriptive or mathematical justification.





In this regard the exponential transit time distribution is first considered briefly, followed by revisiting the L-shaped gamma model of Kirchner at al. (2001). Some comment is then given on computer simulations which might appear to support L-shaped transit time distributions.

## 2 One-parameter exponential distributions

The best known exponential transit time model is the "well-mixed" case where an exponential distribution arises from all tracer particles in a store having equal probability of exiting the store in a small time interval. However, any water store in the physical environment will still require the passage of some amount of time for a tracer pulse to disperse through the store to the observation point, independent of any mixing or partial mixing process. Again, this excludes any tracer particle introduced "at" the observation point because such particles have not undergone store transit to reach the observation point.

Therefore the condition of $f(0) = 0$ will apply and there must still exist at least one mode in the true transit time distribution for some $t > 0$. This applies even though an exponential distribution may give good approximation to a set of transit time data as a whole. See, for example, Fig.7 of Rodhe and Nyberg (1996).

As noted by Leray et al. (2016), theoretical derivations of exponential transit time distributions have also been obtained in a non-mixing context for some idealised aquifer situations (Eriksson, 1958; Haitjema, 1995; Leray et al. 2012; Raats, 1977; Vogel, 1967). However, regardless of the derivation context for the exponential case, the L-shaped issue for small $t$ still remains whereby some tracer particles are required to be present at the observation point at $t = 0$ without ever having entered the store. This leads to awkward descriptors for exponential distributions such as a requirement for "very short" transit times (McGuire and McDonnell, 2006) or "infinitesimal short" transit times (Amin and Campana, 1996).

Such conceptual problems are easily avoided by rethinking exponential transit time distributions as two-parameter gamma or two-parameter Weibull distributions with shape parameters marginally greater than 1 (Fig. 1). This gives the necessary condition of $f(0) = 0$ while still maintaining the exponential form except for $t$ close to zero. However, the situation is not so easily rectified for L-shaped gamma distributions, discussed in the next section.





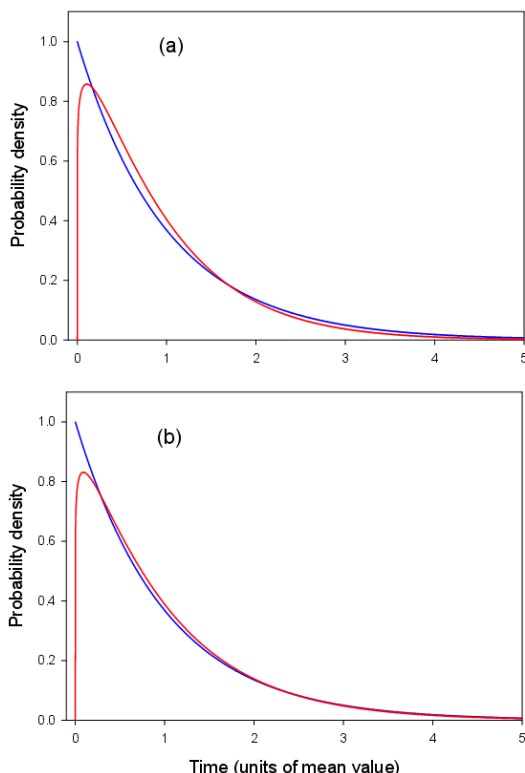

**Figure 1: Exponential distribution comparisons: (a) against a Weibull distribution (b) against a gamma distribution. Exponential distribution is shown in blue; both Weibull and gamma distributions have shape parameter values of 1.1; all distributions have a mean value of 1.0.**

## 3 Two-parameter gamma distributions ($\alpha < 1$)

The L-shaped family of gamma distributions ($\alpha < 1$) was introduced to hydrology by Kirchner at al. (2000) as an empirical choice of transit time distribution possessing power spectrum properties similar to those of a recorded time series of chloride data. A degree of theoretical support for this gamma model was presented by Kirchner at al. (2001), noting that under certain

10 conditions there are similarities between the L-shaped gamma form and an L-shaped transit time distribution obtained from an advection-dispersion model.

The specific advection-dispersion model was for a pulse of tracer added uniformly along a uniform slope, with tracer transported via advection and dispersion to an observation point at the base of the slope at distance $x = 0$. A standardised

15 form of the resulting advection-dispersion transit time distribution $h(\tau)$ is given by Eq. (11) of Kirchner at al. (2001). Unfortunately, a typesetting error omitted a set of brackets and the equation is reproduced here as Eq. (3) with a minor change in symbolism:




$$h(\tau) = (4\pi Pe\,\tau)^{-1/2}\exp[-(1/4)Pe\,\tau][1-\exp(z_0^2 - z_L^2)]$$
$$+ \quad [\mathrm{erf}(z_L) - \mathrm{erf}(z_0)]/4$$

(3)

where

$$z_0 = -(Pe\,\tau)^{1/2}$$
$$z_L = (Pe/\tau)^{1/2} + z_0$$

and $Pe$ is Peclet number and $\tau$ is standardised time.

It is evident from Eq. (1) that $h(\tau) \to \infty$ as $\tau \to 0$.

Eq. (3) was obtained by Kirchner at al. (2001) as an integral over $x$ from $x = 0$ to a ridge crest. However, it is not appropriate to start the integration exactly from $x = 0$ because this would include tracer particles already at the observation point at time $t = 0$. Starting the integration at any $x > 0$ will yield a different transit time probability density function $g(\tau)$, which will have the property $g(0) = 0$.

An equivalent argument can be made by noting the two-parameter inverse Gaussian form of Eq. (8) of Kirchner at al. (2001). Therefore $h(\tau)$ corresponds to an infinite mixture of inverse Gaussian transit time distributions. This infinite mixture distribution can be represented to any degree of accuracy as a finite mixture distribution − in this case a finite mixture of a sequence of inverse Gaussian distributions with progressively decreasing mean and variance as the tracer input point $x_*$ decreases by increments toward the observation point at $x = 0$. A sense of this distribution sequence can be seen in Fig. 3 of Kirchner at al. (2001).

However, the end-member in this transit time distribution sequence, corresponding to $x_* = 0$, is a degenerate distribution with zero variance and all probability density at $\tau = 0$. This is the cause of $h(\tau)$ tending toward infinity as $\tau \to 0$. With the degenerate distribution removed, the resulting transit time finite mixture distribution $r(t)$ will have the property $r(0) = 0$ and not $r(\tau) \to \infty$ as $\tau \to 0$.

The Kirchner et al. (2001) conceptual model can also be viewed as a random walk between a reflecting barrier $R$ (the ridge crest) and an absorbing barrier $A$ (the observation point at $x = 0$), with $x_*$ being equally likely anywhere between the barriers. However, $x_*$ must be restricted to be within the bounds $A < x_* \leq R$ and not $A \leq x_* \leq R$. See, for example, Weesakul (1961). That is, no random walk is initiated from the absorbing barrier because in that case there can be no random walk. So again $f(0) = 0$.





Unlike the case of exponential transit times, L-shaped forms of two-parameter gamma transit time distributions are not so easily mitigated by a slight change of α. This is because best fits to data by gamma distributions, for catchment systems at least, are typically achieved by having gamma α values considerably less than the α = 1 exponential distribution special case (Kirchner et al., 2000; Kirchner et al. 2010, Hrachowitz et al. 2010, Godsey et al. 2010). The issue therefore remains of

$f(0) > 0$, which in this case is actually $f(t) \to \infty$ as $t \to 0$.

One way around the gamma L-shaped problem might be to introduce three-parameter gamma distributions with positive-valued location parameters ω somewhere near zero. However, this would lead to unnatural transit time distributions with first derivative discontinuities at $t = \omega$. A similar issue would arise if using truncated gamma distributions with truncation from below. A better prospect is to seek alternative parametric transit time distributions with $f(0) = 0$ but which permit a

mode near zero. Two-parameter inverse Gaussian distributions or two-parameter lognormal distributions may find application here as alternatives to the gamma distribution. Finite mixtures of such distributions might also be a possibility.

**4 Computer simulations**

Similar issues arise with computer simulations of transit time distributions which might appear to give support to L-shaped

distributions. Simulating the movement of tracer particles by discrete time steps $t = 0, 1, 2 \ldots$ requires an initial state of having no tracer particles at the recording node at time $t = 0$. This is because when $t = 0$ there has not yet been any particle movement simulated anywhere in the store, so the first simulated arrival time can be no earlier than $t = 1$. However, the zero tracer particle frequency at $t = 0$ is still part of the overall simulated transit time distribution.

In this regard it is of interest that Fiori and Russo (2008) report an $R^2$ goodness of fit measure of 0.97 from matching a set of

simulated transit times with a two-parameter gamma distribution with α = 0.77. However, this good match appears to have been achieved by the fitting process ignoring the zero frequency of simulated tracer particles at $t = 0$. The fit measure for an L-shaped gamma distribution would have been much worse if the mismatch at $t = 0$ had been taken into account.

**5 Discussion and conclusion**

It is important to make the distinction between the true transit time distributions of nature and probability density functions which are selected by investigators on the basis of yielding good fits to recorded data. Unlike the thought experiment described earlier, there are no prefect recorders counting arriving tracer particles at some geometrical point. It is unlikely therefore that a null hypothesis of an L-shaped transit time distribution would be rejected with real data, taking into account that recorded time series of tracer output concentrations are complicated by mixing processes varying over time.



The analogy can be made with respect to making transit time distribution inferences from an L-shaped transit time data histogram with declining frequencies $u(1) > u(2) > u(3) \ldots > u(n)$ . In the absence of other information, the obvious parametric estimate of the underlying distribution $f(t)$ would be obtained from fitting an L-shaped probability density function to the data histogram.

However, if it is known independently that $f(0) = 0$ then a unimodal probability density function would be the better choice, with the mode located somewhere within the first bin interval of the histogram. This choice is made on the basis of prior knowledge and not by data-based rejection of a null hypothesis of an L-shaped transit time distribution. Both the unimodal and L-shaped distributions might describe the histogram data with equal accuracy and their cumulative distribution functions could be indistinguishable for practical purposes.

This ability of distributions to mimic each other to some degree suggests that re-analysis of past data using probability density functions other than L-shaped will not necessarily result in different hydrological conclusions.  The argument here has been essentially one of appropriateness. That is, because transit time distributions are characterised by $f(0) = 0$ and one or more modes, it is appropriate that they should be modelled by probability density functions of similar form. However, this is not simply an abstract academic argument because recognition that transit time distributions are not L-shaped opens the
possibility that some catchments suffering a pulse of contaminant input may experience later peaks of contaminant concentrations in stream discharge, as opposed to always declining in concentration over time.

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
