# Peer review of "Technical note: Transit time distributions are not L-shaped"

_Hydrology and Earth System Sciences, 2017_

## Referee Comment (RC1) · Anonymous Referee #1 · 2 Sep 2017

This technical note discusses the general form of pdf employed in transit time distribution analyses. The entire argument is based on the argument that because, on physical grounds, f(t=0) = 0, the frequently-used forms are inherently "wrong" because they have the property that f(t=0) > 0. The author therefore suggests that alternative pdf's should be used, although the author does not offer concrete suggestions.

On a philosophical level, the author is correct – but as the statistician George Box wrote: "Essentially, all models are wrong, but some are useful." So while we are all well aware that the property at t=0 is incorrect, I consider this a minor, irrelevant point. Many existing models "work" well, notwithstanding this point.

In principle, this Technical Note could be published, but quite honestly, I doubt that it will generate much interest. I leave this decision to the Editor. Two points below should

[Figure]

be addressed, however, prior to potential publication.

First, the author uses the term "L-shaped probability density functions", which I find somewhat misleading. Not all pdf's that have $f(t=0) > 0$ – a single point property – look like an "L".

Second, the author might note that not all currently-used pdf's are "L-shaped" – for example, transit time distributions based on the continuous time random walk (CTRW) include the case for $f(t=0) = 0$ — see, e.g., Dentz et al., Transport behavior of coupled continuous-time random walks, Phys. Rev. E, 78, 041110, 2008 (see the coupled case). The CTRW has been applied frequently and successfully in analyses of a range of groundwater applications (e.g., the citations in the Phys. Rev. E paper). Thus, the author's "call" for consideration of "correct" pdf's has in some sense already been heeded.

---

## Author Comment (AC1) · 3 Sep 2017

My thanks to Referee # 1 for comments. Responses follow below.

The referee's second paragraph invokes the argument of practical utility. That is, it is acceptable to knowingly use an incorrect form of transit time distribution provided the outcome is still useful, presumably as measured by fits to data. However, that argument is predicated on the assumption that transit time distributions f(t) with f(0)=0 are consistently less useful. While specific instances might be cited where f(0) = 0 fares less well than L-shaped forms, this by no means constitutes a proof of the assumption. Distributions are flexible entities and, for example, it will always be possible to find a finite mixture of f(0) = 0 distributions which result in data matching as good or better than any L-shaped distribution.

Given an absence of a practical utility argument, the question then arises as to why we should knowingly employ transit time distributions of incorrect form. That is, why would we use L-shaped transit time distributions which have no mode when all transit time distributions must have at least one mode? The referee takes the view that this is a minor and irrelevant point. On the contrary, we know little enough about the forms of transit time distributions and it would seem only sensible to make as much use as we can of our knowledge that f(0) = 0, and not simply ignore it.

With respect to the two specific points which the reviewer wished addressed:

(i) The definition of L-shaped distributions was f'(t) < 0 and not f(0) > 0. L-shaped forms are always obtained from f'(t) < 0 for all  $t \ge 0$ .

(ii) There was never any intention in the paper to imply that only L-shaped probability density functions have been used through the history of transit time modelling. This point can be noted while using the additional references mentioned by the referee. The use of f(0) = 0 probability density functions to represent transit time distributions extends at least back to Turner and Barnes (1998), cited by Kirchner et al (2000) as an example of utilising the gamma distribution with shape parameter > 1. However, L-shaped distributions also continue to be used. Previous use of f(0) = 0 distributions is therefore of no consequence to the message of the paper – that L-shaped distributions do not properly reflect the form of transit time distributions and the call is made for their use to be discontinued.

Further responses follow to specific points mentioned briefly by the referee.

It is beyond the scope of this brief technical note to extend to advocating the merits of any specific alternative transit time distribution. The lognormal distribution is mentioned as an empirical possibility. So too is the inverse Gaussian distribution (which is the transit time distribution for Brownian motion with drift). Mention might also be made of the CTRW model noted by the referee.

With respect to whether the paper will generate interest is not for me to say. Perhaps the referee is right in contending that the paper would raise little interest. However, my feeling is that there is potential for hydrological interest. Specifically, given that f(0) = 0 it follows that all transit time distributions have a lag time to first mode. This lag time (which could be made dimensionless by scaling to the median) might be so close to zero that it is irrelevant. On the other hand, there might be possibility of some degree of displacement of the first mode from time zero. This would raise questions concerning the physical processes creating the lag. Similarly, the possibility of multiple modes and their physical associations might be raised from some data sets. Such issues could not be contemplated in the context of L-shaped distributions. So to that extent the paper has scope for opening the way to better hydrological questions.

Turner, J. V., Barnes, C. J. *Modeling of Isotope and Hydrogeochemical Responses in Catchment Hydrology* IN: Isotope Tracers in Catchment Hydrology (eds Kendall, C., McDonnell, J. J.) 723-760 (Elsevier, Amsterdam, 1998).

Kirchner, J.W., Feng, X.H., Neal, C. (2000) Fractal stream chemistry and its implications for contaminant transport in catchments, Nature, 403, 524–527.

---

## Referee Comment (RC2) · Anonymous Referee #2 · 7 Sep 2017

The note of Earl Bardsley raises the valid point that water parcels and solute particles cannot traverse a catchment with infinite velocity, implying that the probability density function $p(\tau)$ of travel time $\tau$ should start at zero. The author likes to call distributions with non-zero probability density at time zero as "L-shaped" and argues with modes of the distributions at $t > 0$. This is not only confusing but can also be wrong. You could think of a pdf $p(\tau)$ that starts of at a finite value, increases, and then drops. This distribution does not look like an L but would be conceptually wrong nonetheless. The physical constraint that the author should enforce is $p(0) = 0$. This can be stated more or less in a single sentence without the thought experiment leading to equations 1 and 2, which was only partially enlighting.

So far, so good. But is this really new? I don't think so. And it misses the point what

we are actually doing when we fit models. Simple parametric distribution are chosen by practioners (and reseacrhers) in hydrology because of parsimony rather than correctness. The exponential distribution is the maxentropic distribution for a non-negative variable with known mean. For any other distribution, you need more information in the data. Often, travel-time and residence-time distributions occur in a wrapped way, e.g., when assuming a transient-storage zone in stream transport or when assuming kinetic sorption (, where it is the time that a solute stays in the immobile zone rather than the passage through the domain), thus, travel-time distributions may be convoluted with other distributions (in the simplest case the input signal) to obtain the measureable quantity (i.e. a breakthrough curve in the outlet of the system). Under such conditions, the data are often not good enough to fit anything better than an exponential distribution. Does this mean that a serious researcher really believes that this the correct distribution? Hopefully not! It simply implies that you were not able estimate more than the mean of the distribution, and you take the simplest model to express that. Any interpetation beyond that is pure hydrofantasy. If you fit the inverse-Gaussian distribution, as suggested by the author, you fit the analytical solution of the advection-dispersion equation with constant coefficients; but you'd better not believe that transport is Fickian *in reality* (most likely the tails are wrong), you actually fit the mean and variance of the distribution. That's often OK, but it already requires two parmeters rather than one and thus more information than the exponential model.

As a way out, the author suggests to use the gamma distribution with a shape parameter slightly bigger than one. That enforces the pdf to start at zero, but the author cannot really say how much bigger than one the shape parameter should be. Hence, the suggestion is not truly very helpful.

Thus, if you have really good data to calibrate a model, you can choose from a large variety of parametric travel-time distributions that start off at zero (we often use a generalized inverse Gaussian function with an offset; it has no real physical meaning but does a good job in tracer tests). If the data are extremely good, you can determine the

travel-time distribution by non-parametric deconvolution, enforcing non-negativity and a starting-value of zero (we have done that, too). But if your data are not that great, you stick to the simplest distributions thinkable. This does not imply that you believe the distribution to be exact, but it may have the one feature that you can extract from the data. If there is any plea to be made than that we should not overinterpret fitted model parameters.

On an editorial side: Don't write about "L-shaped" distributions, and don't write about "stores" (which should be "storage zones" to avoid confusion with shops). And of course, it does make sense to integrate from the hill crest to the river if you want to derive an analytical expression (by the way, rain can even fall into the river itself).

---

## Author Comment (AC2) · 8 Sep 2017

My thanks to Referee # 2 for comments. Responses follow below.

" The author likes to call distributions with non-zero probability density at time zero as "L-shaped" and argues with modes of the distributions at t > 0."

It is not a question of "likes to call". The term "L-shaped" is a widely used as a distribution descriptive term. For the purposes of the paper a specific definition is used. That is, a transit time distribution f(t) with origin at zero is defined to be L-shaped if f'(t) < 0 for  $t \ge 0$ . I could not imagine anyone contemplating defining a distribution to be L-shaped on the basis of a non-zero probability density at time zero. I am totally perplexed therefore why I should stand accused by both reviewers of doing just that. After all, f'(t) < 0 for  $t \ge 0$  is hard to misinterpret.

Of course there are other distributions forms which are also conceptually incorrect. However L-shaped forms (as defined in the paper) are widely used as transit time distributions and focus therefore is only on those forms – as indicated by the title of the paper.

I am entirely happy to remove the thought experiment in the final version of the paper, if accepted. It was presented in the Discussion version in anticipation of the kinds of criticism that would be raised.

The referee's second paragraph raises the important points of parsimony and distribution identifiability. At first glance it all sounds plausible. Assume a simple L-shaped distribution, such as the 1-parameter exponential. This is then applied to data of complex origin and, assuming a satisfactory fit is obtained, it would be reported as "a satisfactory data fit was obtained using an exponential transit time distribution". Similar studies over the years would build up to "numerous studies have shown that the exponential distribution is a suitable model for many transit time distributions", followed by the obligatory extended citation list.

However, knowingly using an incorrect distribution form for convenience carries its own penalty against good hydrological science. That is, using an exponential distribution and doing nothing more ignores the fact that the transit time distribution must in reality have a mode at some t > 0. As noted by the referee, distribution identification is essentially impossible with real-world data. That cuts both ways. If the exponential distribution is recognised as never being correct for time zero, then that should raise the question "how far away from time zero might the first mode be located while still leading to an equally good fit to data?". This is a legitimate hydrological question and it does not amount to the impossible task of identification of a specific alternative transit time distribution. For example, a nonparametric histogram might be utilised as alternative. Such a distribution could be constrained to have the correct form (zero probability near zero) and would also provide a degree of sensitivity measure for the transit time distribution application concerned.

The reference in the paper to the possible use of unimodal two-parameter distributions was not in terms of more complex alternatives to the 1-parameter exponential distribution, but as potential alternatives to the 2-parameter gamma distribution with shape parameter < 0. As noted in the paper, the exponential distribution has been shown to be unsuitable as a transit time distribution in a number of catchment studies. And yes, there is no implication of Fickian systems in reality if using inverse Gaussian distributions – just as Kirchner et al (2001) presumably did not consider they were modelling Fickian slope water movement when they used a mixture of inverse Gaussian transit time distributions.

**"As a way out, the author suggests to use the gamma distribution with a shape parameter slightly bigger than one."**

No – in fact for the purposes of using a single parametric distribution where the exponential distribution gives suitable fit to data, the point is made in the paper that there no need to seek a "way out". As mentioned above, the more important issue is the extent to which the data might permit the mode to be displaced from zero, even when the exponential model works well with the data. However, for the purposes of using the exponential distribution as a single parametric distribution which works with the data, there is no need to consider some specific numerical value of a shape parameter marginally greater than 1 because the "L-shaped" and "non-L-shaped" versions will be indistinguishable in this case (unlike gamma distributions with shape parameter considerably less than 1). To that extent the "call" for abandoning L-shaped distributions is moderated for the special case of the exponential distribution.

**" But if your data are not that great, you stick to the simplest distributions thinkable."**

Outside of quantum concepts, the true nature of the physical world exists entirely independently of our ability to measure it. Data quality is therefore no excuse to use a model which we know to be incorrect in some specific aspect, whether the model concerned be simple or complex. Transit time distributions with zero probability density at time zero always exist in nature even if we have no measurements at all. To be sure this is limited knowledge, but it would be unfortunate to ignore it. Of course, with poor data quality any attempt at parameter interpretation will not mean very much.

With respect to the "editorial" comments:

I would not drop the "L-shaped" title because that describes exactly the content of the paper, and just what is meant by L-shaped is exactly defined in the context of the paper, as mentioned earlier.

Point taken with respect to water "stores", though I would prefer something like "water flow system".

It does not make sense to integrate exactly to the river system because that includes particles already at the observation point at time zero and are therefore not part of the transit time distribution. That is why it was necessary for this discussion document to start off with the thought experiment description. With respect to the "rain falling into the river" – if the tracer particles concerned fell into the river anywhere upstream of the conceptual observation point then the particles would require finite time to be carried to the observation point. If the rain falls directly "on" the conceptual recorder then the particles have not been involved in transiting the flow system and are therefore not part of the transit time distribution.

Finally, a response to the referee query "is this new?". In some ways this is a hard question to answer. Perhaps it really is common knowledge that L-shaped transit time distributions cannot exist in reality. If that is the case then it does seem unusual then that, for example, gamma distributions with shape parameters less than 1 should be utilised in transit time modelling. Given that two parameters are required, it would seem preferable to make a thorough search through the many available 2-parameter distributions with zero probability density at time zero to see if the data can be matched as well. To my knowledge there has not been a previous paper drawing attention to the impossibility of L-shaped transit time distributions, nor any call to seek to find how far from time zero the data might permit the first transit time mode to be located.

---

## Referee Comment (RC3) · Anonymous Referee #3 · 18 Sep 2017

This technical note has one single subject; that the authors is dissatisfied with travel time distributions which have a nonzero value at time t=0 (the moment of tracer application). The authors seems to use a causality argument - travel velocities are finite, and therefore not a single tracer particle could have reached the outlet in no time. However, the travel time distributions, expressed as probability density functions, of Kirchner et al. (2001) which he criticizes explicitly cover the case of spatially distributed tracers, e.g. natural tracers in rainfall or a homogeneous sprinkler system. Thus, there are tracer particles which are precisely at the outflow at t=0, which do not disperse, leading to sharp peak, and others further away which do disperse and lead to long tails in the distribution which is the central focus of the Kirchner et al. work. The author seems to be uneasy with divergent pdf's. They are rather common and not problematic as

long as the singularity is not too strong - more precisely, if the integral over any finite time interval is always finite. In one dimension, this it the case when the singularity is weaker than $1/t$ - e.g. the $1/t^{0.5}$ of eq. 11 in the Kirchner et al. article. The reviewer cannot see that the printed equation (11) contains an error with brackets - there could have been an extra pair of brackets around the argument of the second exponential, but this is optional and does not change the equation. The current author uses units where the mean travel time, $\tau_0$, is set to 1 (why?), but then there is a factor 1/2 missing in the definition of $z_0$. This is an error, but since he does not develop that further, it is overall not an important one. Unfortunately, the author does not provide a framework to mitigate the "problem" of $f(0) > 0$ (if this is a problem at all). Arbitrary changing integration limits can't do the trick. He also does not demonstrate (analytical or, if not easily possible, numerically) that whenever $A < x^*$ (line 24 on page 5), then $f(0) = 0$. The problems we have in evaluating and interpreting tracer studies are not here. One has to tackle instationarities, differences in hydraulic conductivity, imperfect recovery (e.g. detection limits at high dilution) etc., which is a can of worms in many cases. Honorable approaches like the CXTFIT software are restricted to 1D, whereas Hydrus-3D suffers from too many parameters. It is here where we should put our emphasis in tracer hydrology; the "problem" discussed in this contribution is not significant and does not lead to problems with causality or ill-definedness of the expresssions obtained. The primitive exponential distribution is for demonstration purposes only - like the harmonic oscillator in physics - and is to be found in textbooks and reviews on tracer hydrology, but the research field is far beyond this. For pointwise tracer injections away from the stream / outlet with your detector, you will have $f(0) = 0$, but this is not what is discussed in Kirchner et al. (2001).

---

## Referee Comment (RC4) · Anonymous Referee #4 · 19 Sep 2017

The purpose of this contribution is to assert that probability distributions for water transit time through hydrologic systems cannot be monotonically decreasing. The author calls distributions like this "L-shaped." He demonstrates through a thought experiment and discussion of commonly used distributions why water that was present in the storage cannot also leave the storage at a time of zero. Overall, the argument makes sense, but I am not sure I really understand the contribution of the manuscript given that this issue has been recognized in the literature and even the author claims that a more correct form of the distribution may not result in different hydrological conclusions. Technically, the author's primary point about the incorrectness of the early-time behavior of the transit time probability distribution is fine, especially for stationary distributions of a generally smoothed shape. However, I don't think fitting theoretical distributions

where f(0) = 0 is more informative or helpful hydrologically, particularly if the distribution maxima is close to t = 0. The author recognizes the challenge of selecting distributions based on fitting as being problematic, but moreover, he fails to extend his analysis to understanding the actual shape of transit time distributions. There is now a significant literature illustrating how complex these distributions are without making any assumption of distribution shape (e.g., van der Velde et al. 2012, 2015, Benettin et al. 2015ab, Birkel et al. 2015, Kim et al. 2016). They are dynamic in time and perhaps unique on any given time (e.g., Duffy 2010, Rinaldo et al. 2015). Again, this short technical note is not incorrect, but I don't know how valuable or impactful it will be, particularly without recognizing the recent literature on dynamic transit time distributions and the fact that actual transit times may not be smooth shaped functions or parametrically characterized.

Specific comments: I spent some time Googling "L-shaped" probability distributions because I found it to be an odd descriptor of a probability distribution. It is used some, but it is not very common and if impact and attracting readers is a goal (and it should be), then I would consider revising the title. Maybe "transit time distributions with an early-time probability maxima" would work? I actually cannot think of good title, but I did immediately question the term "L-shaped" distributions.

P1, L19: insert often before assumed

P2, L7: delete the second "and"

P2, L12: Yes, but at f(0), the probability could also simply be low (i.e., not zero) particularly if there are some parcels of water with almost instantaneous exit like those falling on the recorder.

P3, L10: change from "have not undergone store transit" to undergone transit through the storage.

P3, L14: Missing reference, but I think the author is referring to Rodhe et al. (1996).
P5, L7: should be referring to Eq. 3.

P6, L9-10: A real problem here is that the author is ignoring a significant body of work and much of the latest work on catchment transit time where the community seems to be avoiding parametric distributions in the first place. A good example is the storage selection approach (e.g., see Rinaldo et al. 2015). There are fewer and fewer papers in the literature using parametric, stationary distributions.

P6, L19: comma after "regard"

P6, L21-23: It might be helpful to reference a few studies that have found transit time distributions similar to those the author describes. For example Rodhe et al. (1996), Simic and Destouni (1999), and Davies et al. (2013) from the Gårdsjön Catchment and McGuire et al. (2007), Fiori and Russo (2008), Dunn et al. (2010), and Darracq et al. (2010) show distributions as described by the author. Furthermore, many of the studies that show transit time distributions from dynamic models, do not assume L-shaped distributions (or any functional form for that matter).

P6, L7: Is there a question of time step or bin interval? It seems to me that for very fine bin intervals, it is possible to have transit time pdf with a mode very near t = 0 and hydrologically this will not result in transport that this very different from an L-shaped distribution. The other issue is that often continuous pdfs of transit time are actually implemented discretely in time such that for times near zero, they may be indistinguishable from L-shaped distributions like a gamma with alpha < 1.

References:

Benettin, P., S. W. Bailey, J. L. Campbell, M. B. Green, A. Rinaldo, G. E. Likens, K. J. McGuire, and G. Botter (2015a), Linking water age and solute dynamics in streamflow at the Hubbard Brook Experimental Forest, NH, USA, Water Resour. Res., 51, 9256–9272, doi:10.1002/2015WR017552.

Benettin, P., J. W. Kirchner, A. Rinaldo, and G. Botter (2015b), Modeling chloride transport using travel time distributions at Plynlimon, Wales,Water Resour. Res., 51, 3259–3276, doi:10.1002/2014WR016600.

Birkel, C., Soulsby, C., & Tetzlaff, D. (2015). Conceptual modelling to assess how the interplay of hydrological connectivity, catchment storage and tracer dynamics controls nonstationary water age estimates.ÂăHydrological Processes,Âă29(13), 2956-2969.

Darracq, A., Destouni, G., Persson, K., Prieto, C., & Jarsjö, J. (2010). Quantification of advective solute travel times and mass transport through hydrological catchments.ÂăEnvironmental Fluid Mechanics,Âă10(1), 103-120.

Davies, J., Beven, K., Rodhe, A., Nyberg, L., & Bishop, K. (2013). Integrated modeling of flow and residence times at the catchment scale with multiple interacting pathways.ÂăWater Resources Research,Âă49(8), 4738-4750.

Duffy, C. J. (2010). Dynamical modelling of concentration–age–discharge in watersheds.ÂăHydrological processes,Âă24(12), 1711-1718.

Dunn, S. M., Birkel, C., Tetzlaff, D., & Soulsby, C. (2010). Transit time distributions of a conceptual model: their characteristics and sensitivities.ÂăHydrological processes,Âă24(12), 1719-1729.

Kim, M., L. A. Pangle, C. Cardoso, M. Lora, T. H. M. Volkmann, Y. Wang, C. J. Harman, and P. A. Troch (2016), Transit time distributions and StorAge Selection functions in a sloping soil lysimeter with time-varying flow paths: Direct observation of internal and external transport variability, Water Resour. Res., 52, 7105–7129, doi:10.1002/2016WR018620.

McGuire, K. J., Weiler, M., & McDonnell, J. J. (2007). Integrating tracer experiments with modeling to assess runoff processes and water transit times.ÂăAdvances in Water Resources,Âă30(4), 824-837.

Rinaldo, A., P. Benettin, C. J. Harman, M. Hrachowitz, K. J. McGuire, Y. van der Velde, E. Bertuzzo, and G. Botter (2015), Storage selection functions: A coherent framework

for quantifying how catchments store and release water and solutes, Water Resour. Res., 51, 4840–4847, doi:10.1002/ 2015WR017273.

Rodhe, A., L. Nyberg, and K. Bishop (1996), Transit times for water in a small till catchment from a step shift in the oxygen 18 content of the water input, Water Resour. Res., 32(12), 3497–3511.

Simic, E., & Destouni, G. (1999). Water and solute residence times in a catchment: Stochastic‐mechanistic model interpretation of 18O transport.ÂăWater Resources Research,Âă35(7), 2109-2119.

van der Velde, Y., Torfs, P. J. J. F., Zee, S. T., & Uijlenhoet, R. (2012). Quantifying catchment‐scale mixing and its effect on time‐varying travel time distributions.ÂăWater Resources Research,Âă48(6).

van der Velde, Y,ÂăHeidbüchel, I,ÂăLyon, SW,ÂăNyberg, L,ÂăRodhe, A,ÂăBishop, K, andÂăTroch, PAÂă(2015),ÂăConsequences of mixing assumptions for time-variable travel time distributions.ÂăHydrol. Process.,Âă29,Âă3460–3474. doi:Âă10.1002/hyp.10372.

---

## Author Comment (AC3) · 20 Sep 2017

My thanks to Referee 3 for comments – responses follow below.

Firstly, with reference to the Kirchner et al (2001, Eq 11), I did indeed omit the 0.5 multiplier in the definition of z0. My thanks for picking up my poor copying of the expression – although as noted this has no implications for the argument in the current paper.

I guess my reference to a "typesetting error" may have been a little too strong. And there was never any suggestion of a mathematical error. I think though the expression  $1/4Pe(\tau/\tau_0)$  would be clearer with a second set of brackets. If the current paper is accepted I would reproduce the original equation with the extra brackets added but without reference to a typesetting error.

In the Kirchner et al (2001), standardised time was represented as  $\tau/\tau_0$ . It seems more compact to simply define  $\tau$  to be standardised time without having the explicit rescaling. If this is seen as an issue then there is no problem to revert back to  $\tau/\tau_0$ .

There is no disagreement that an arbitrary change of integration limits would be just that – arbitrary – and therefore not very helpful.

An argument is requested to demonstrate that if  $A < x^*$  then f(0) = 0. The argument is simple. The distance  $x^*$ - A is some interval of distance greater than zero. A particle which starts its random walk from point  $x^*$  at time t = 0 cannot also arrive at A at time t = 0, unless it travels at infinite speed. Therefore f(0) = 0 if  $A < x^*$ .

I have no reason to have any issue with singularities in probability density functions as such. But there is an issue if the singularity creates an L-shaped transit time distribution, which must be an incorrect form.

Further to this, it is evident from paragraphs 1 and 2 of Referee #1 that, apparently, everyone is very aware than anything other than f(0) = 0 is incorrect, although Referee #1 felt that this truth was of no particular consequence. In contrast, Referee #3 maintains:

"Thus, there are tracer particles which are precisely at the outflow at t=0, which do not disperse, leading to sharp peak, and others further away which do disperse and lead to long tails...".

So who is correct? Is it Referee #1, or Referee #3? If Referee #1 is correct then the tracer particles in the transit time distribution all transit some non-zero time to an observation point and Eq (1) in the present paper applies. If Referee # 3 is correct the tracer particles of the transit time distribution are comprised of two sub-populations : those which transit and those which do not. In that case Eq (2) of the present paper would apply.

It is of course Referee #1 who is correct, because particles which do not transit cannot be a subpopulation of a transit time distribution.

The current paper does indeed have a single subject, as noted in the title - drawing attention to the fact that transit time distributions cannot be L-shaped. And, by implication, noting that any mathematical derivation leading to L-shaped transit time forms must contain some degree of error. Utilising the property f(0) = 0 is simply a means on the way to achieve that end.

It is beyond the scope of this brief technical note to use the rejection of L-shaped forms to provide a starting point for alternative approaches which take into account data gathering and the physical complexities of catchment processes. However, elsewhere in HESSD I do have a paper morphing its way through several iterations which may or may not turn out to be helpful for nonparametric approaches.

Finally, it is worth repeating the comment at the end of the paper. The argument is not simply variations on some obscure theme as to how many tracer particles might dance on the head of a needle. The assumption of L-shaped transit time distributions is in fact potentially risky in practice. That is, it is entirely possible that a brief pulse of contaminant widespread over at catchment could result in multiple peaks of downstream contaminant concentrations, and not just decline consistently over time.

My thanks again to Referee # 3. I understand there is also a Referee # 5 due to release comment, so I will perhaps wait and respond to Referees #4 and #5 together.

---

## Referee Comment (RC5) · Anonymous Referee #5 · 28 Sep 2017

The technical note focusses at a single point that in a pdf describing a breakthrough experiment in an catchment f(t=0) must be zero by necessity. Distributions fitted to breakthrough curves not fulfilling this criterion are termed "L-shaped" by the author. He proposes to avoid them.

The argument would benefit from having its physical aspects more clearly delineated from its mathematical aspects. In many cases the problematic assumption is stationarity as a prerequisite for using pdfs for describing breakthrough curves in the first place. Furthermore, does the argument still hold when mathematical descriptions of what is termed "store" employ fractal dimensions?

In my opinion, the author does not provide a convincing argument that the situation is properly characterized as an impossibility. I propose using "inconsistency" within an

implicitly given physical model. It can be quite hard to show that something is impossible within a given formal framework. The part of hydrology addressed by this technical does not have such an accepted theory, tracer hydrology includes many pragmatic approaches.

The most unclear point to me is the idealized catchment employed in the thought argument. The argument by the author seems to assign a physical storage to this ideal catchment with definite physical features. In the further discussion the properties of the storages are lifted to an 'ontic level' ("the true situation"). A number of questions occur: Is the idealized catchment on Earth? Then it will inevitably contain biota interacting with the forms of transport pathways. Is the idealized catchment in a unique state (by experimental preparation) or taken out of natural history?

One could envision a second thought experiment in which the boundary of a natural catchment containing biota is entirely defined by observations of fluxes across them. Its external relations are described in classes of behavioral equivalence (similar to theoretical descriptions of computer programs) and any assignment of local internal properties becomes an over-interpretation of the available data. And of course, as already brought up in the ongoing discussion, input fluxes may hit the boundary producing output.

Summarizing I do not find this argument very compelling or a helpful focus for tracer studies.

specific comments: I could not find an error in (11).

---

## Short Comment (SC1) · 28 Sep 2017

Re: The zero–time–to–peak characteristic of a probability density function

I'd like to respond, at this late stage of open discussion period, to the clarion call from the author to banish L–shaped probability density functions from transit time modelling (see Abstract, last sentence).

The initial non–zero probability of a tracer transit time distribution is analogous to the zero–time–to–peak characteristic of a synthetic unit hydrograph.

From the latter's perspective, the L shape of a probability density function (PDF) is a logical consequence of a control volume being considered conceptually a spatially lumped catchment.

To confront, overcome or correct this inherent characteristic or shortcoming of a **single** conceptual store (or reservoir, tank, bucket, tub, etc.), I had resorted to the use of a classical concept relating an S–curve hydrograph and its instantaneous unit hydrograph (Ding, 1974, 2011).

Statistically, the former is a cumulative probability function (CPF), $F(t)$, e.g. Ding (2011, Fig. 3); the latter a PDF, $f(t)$, e.g. Ding (1974, Fig. 1); and that the PDF is the derivative of the CPF, $f(t) = dF(t)/dt$.

For a unit hydrograph, this produces an input–dependent or variable characteristic time, i.e. the higher the rainfall–excess intensity, the shorter the time of the peak ordinate (Ding, 1974, Fig. 3).

References

Ding, J. Y., 1974. Variable unit hydrograph. Journal of Hydrology, 22(1-2), pp.53-69.
Ding, J. Y., 2011. A measure of watershed nonlinearity: interpreting a variable instantaneous unit hydrograph model on two vastly different sized watersheds, Hydrol. Earth Syst. Sci., 15, 405-423, https://doi.org/10.5194/hess-15-405-2011.

---

## Author Comment (AC4) · 8 Oct 2017

My thanks to Referee #4 for taking time to carefully work through the manuscript, and the many constructive comments and reference suggestions.

I do not believe I am much at odds with the comments overall, but offer some responses on specific aspects below.

**... but I am not sure I really understand the contribution of the manuscript given that this issue has been recognized in the literature and even the author claims that a more correct form of the distribution may not result in different hydrological conclusions.**

The Referee comment here about "recognized in the literature" is of interest. Referee #1 also notes that there is widespread awareness of f(0)>0 being incorrect and Referee #2 points out that f(0) = 0 is not really new. It should follow then that there is widespread awareness by authors of the incorrectness of any analytical derivation they might publish which leads to L-shaped transit time distributions. However, I know of no published paper which presents an L-shaped transit time distribution where the mathematical expression is followed by a statement to the effect that "this analytical solution is incorrect for small t because everyone knows that f(0) = 0."

There is actually no grey area here. It is either possible or it is not possible to derive L-shaped transit time distributions via analytical argument. I would certainly hope that that HESS agrees with the latter. However, it is also not really desirable for the submitted paper to be entirely negative in tone, however correct it may be. By way of inserting a more positive note, some alternative distributions are tentatively suggested later as part of the further responses to Referee #4. Hopefully these will go some way to offset the criticisms of Referee #2 and Referee #3 about the lack of suggested alternatives.

Considering now the above Reviewer #4 comment with respect unchanged conclusions. It is entirely possible, perhaps even expected, that use of distributions of correct form will result in no change in hydrological conclusions. However, this is not an argument for accepting continued use of L-shaped distributions.

The reasons are twofold. Firstly, given equal degree of matching to data, it is better science to utilise a distribution form which is not in contradiction to what we know to be the true form of the early time portion of transit time distributions. After all, why would we knowingly choose an incorrect form? The philosophy is well summarised by J.F.C. Kingman in his comment contribution following Folks and Chhikara (1978):

"Although it is often possible to justify the use of a distribution empirically, simply because it appears to fit the data, it is more satisfactory if the structure of the distribution reflects plausible features of the underlying mechanism."

Secondly, continued use of the L-shaped transit time distribution concept is not desirable from the hydrological viewpoint because there are some potentially useful questions which do not get to be asked. In particular, given that f(t) = 0, what is the lag time to first mode? (This assumes that transit time distributions are "smooth" to the extent that the first and second derivatives are continuous so modes are defined.) Is the mode always so close to zero that the lag time can be neglected, or could the lag time be of practical importance? Perhaps there is a distribution of times to first mode, reflecting the time-varying nature of transit time distributions. The mean and variance of such a distribution might be of value as descriptors of the early-time behaviour of transit time distributions from a given hydrological system. It may happen that mixing processes often allow so little distribution information to pass through the system that such questions will seldom be answered satisfactorily. However, if transit time characterisations are so poor, can we be so sure that the first mode is in fact very close to zero? Put another way, how large might a model's lag time be before data fit starts to fall away? Perhaps looking at tracer outputs from physical catchment model results could be of help here.

**Technically, the author's primary point about the incorrectness of the early-time behavior of the transit time probability distribution is fine, especially for stationary distributions of a generally smoothed shape.**

I doubt that it would ever be possible to have stationary transit time distributions in reality, but the stationary or non-stationary nature of tracer transit time distributions is of no consequence to the discussion. We can conceptualise a tracer transit time distribution being defined from an instantaneous uniform spatial scattering of tracer into a catchment, and recording the exit time of the particles as they depart. The experiment might then be repeated at any later time (including immediately after), which would result in a different tracer transit time distributions would have the property f(0) = 0, which in both cases would negate the

possibility of L-shaped transit time distributions as defined here. "Smoothed" shapes are not a strict requirement for defining L-shaped distributions as referenced in the paper. The first derivative must be always negative for  $t \ge 0$ , but it could be discontinuous.

**... but moreover, he fails to extend his analysis to understanding the actual shape of transit time distributions.**

This is far beyond the scope of a short technical note with an argument themed against L-shaped transit time distributions. The paper makes no pretence to suggest analysis leading to deeper understanding of transit time distributions forms as we move further away from t = 0. The Referee is no doubt in a much better position to contribute to that.

Again, this short technical note is not incorrect, but I don't know how valuable or impactful it will be, particularly without recognizing the recent literature on dynamic transit time distributions and the fact that actual transit times may not be smooth shaped functions or parametrically characterized.

Dynamic transit time distributions are something of a distraction because whether tracer transit time distributions are dynamic or otherwise is not of relevance for an argument against L-shaped transit time distributions. All that matters is that all transit time distributions must have some form, and an L-shaped form is not one of the available options. The paper is only concerned with an argument against the use of L-shaped forms as defined in the paper. Fractal situations and the like are outside the scope of the paper.

The point is noted, however, with respect to impact and relevance. This is linked to the need to make some contribution toward alternatives. So while the main message of the paper is directed against use of L-shaped distributions, a suggestion is now made here of alternatives. That is, 1- or 2-parameter distributions of correct form with f(0) = 0 which might serve to replace L-shaped transit time distributions with one or two parameters. There is obviously no specific "best" replacement distribution that can be readily defended. However, it is suggested that the inverse Gaussian distribution should be given consideration as alternative for both 1- and 2-parameter L-shaped distributions, discussed further now.

The inverse Gaussian distribution has origin as a first passage time distribution and hence has f(0) = 0. It has some history as a transit time distribution in hydrology, dating back to the inverse Gaussian mixture model of Kirchner et al (2001), though it was not mentioned explicitly as such in that paper.

The inverse Gaussian distribution can be written in various parameterisations, the utilised form here is in terms of its mean  $\mu$  and shape parameter  $\phi$ :

$$f(t) = \left(\frac{\mu\phi}{2\pi t^3}\right)^{1/2} e^{\phi} \exp\left(\frac{1}{2}\phi\left(\frac{t}{\mu} + \frac{\mu}{t}\right)\right)$$
(1)

Considering first an alternative to exponential transit time distributions, a specific inverse Gaussian distribution is suggested where the shape parameter is fixed at 1. This gives the 1-parameter probability density function:

$$g(t) = \left(\frac{\mu}{2\pi t^3}\right)^{1/2} \exp\left(1 - \frac{1}{2}\left(\frac{t}{\mu} + \frac{\mu}{t}\right)\right)$$
(2)

This distribution has similarities to the exponential distribution except for t near zero (Fig 1). It could be useful in fact to have Fig. 1 here replace Fig. 1 in the submitted paper.

Figure 1: Comparison between the exponential distribution and the inverse Gaussian distribution as given by Eq. (2). Both distributions have mean value = 1.0.

Referee #2 suggests that when we only have confidence to estimate the mean of the distribution, then we should take that mean as being the mean of an exponential distribution. However, the exponential distribution and g(t) here are both defined only by their mean values. Given the choice between single parameter distributions of correct and incorrect forms, it would seem better to select the former when other things are equal.

The other L-shaped distribution considered in the submitted paper is that given by Eq. 11 of Kirchner et al (2001). That distribution was obtained as the additive effect of a multitude of inverse Gaussian distributions with distribution origins distributed uniformly along a slope leading down to a river channel. This gives an L-shaped mixed inverse Gaussian distribution with the river acting as an absorbing barrier. This mixture distribution was referenced by Kirchner et al (2001) as giving some degree of support for L-shaped gamma transit time distributions. As it happens, Eq. 11 of Kirchner et al (2001) is an incorrect analytical expression for transit time distributions because the derivation incorporates particles already at the observation point at time zero, and are not part of the transit time distributions. Its mathematical expression cannot therefore be used as support for L-shaped gamma transit time distributions.

As noted by Reviewer # 3, a focus of the Kirchner et al (2001) paper was also for giving an explanatory model of long-tailing behaviour. The explanation was essentially topographical, with those transit time distributions originating further up the slope segment creating an extended right tail of the inverse Gaussian mixture distribution, as illustrated in Fig. 3 of that paper.

An alternative highly simplified model is proposed here, with some limited degree of hydrological connection. It enables long-tailing to be obtained via a single inverse Gaussian transit time distribution, avoiding L-shaped forms. The simple conceptual model is described below.

A tracer particle is deposited at some point on a slope segment and starts a random walk toward a river channel, which is some finite distance away and acts as an absorbing barrier. Whenever the particle is on the land surface there is a probability p that it will make an incremental movement down the slope toward the stream. There is also a probability q that it will instead make an incremental vertical movement downward from the surface into the subsurface beneath its current surface location. Whenever the particle is in the subsurface there is probability p that it will make an incremental vertically upward and probability q that it will make an incremental movement vertically upward and probability q that it will make an incremental movement vertically upward and probability q that it will make an incremental movement vertically upward and probability q that it will make an incremental movement vertically upward and probability q that it will make an incremental movement vertically upward and probability q that it will make an incremental movement vertically upward and probability q that it will make an incremental movement vertically upward and probability q that it will make an incremental movement vertically downward. Define p + q = 1, so the particle is always moving.

It is recognised that this model does contain multiple hydrological disconnects: horizontal movement on the ground surface is the same as vertical movement in the subsurface, a particle can only return to the ground surface at the point where it last entered the subsurface, and no allowance is made for instantaneous multiple starting points along the slope as in the Kirchner et al (2001) model. However, for all its shortcomings the model yields topographically independent long tailing without needing L-shaped transit time forms. It is introduced here only in the spirit of suggesting an alternative to the L-shaped transit time form derived in Kirchner et al (2001). There is no suggestion that it offers in any way a better approximation to actual catchment processes. In this regard, Scher et al (2002) give a detailed discussion of a continuous time random walk model leading to long-tailing of catchment transit time distributions.

For p > q the model described above is analogous to Brownian motion with drift and the transit time distribution to the channel is an inverse Gaussian distribution. However, as noted in the response to Referee #2, there is no implication that particle movement is Fickian in reality. Presumably Kirchner et al (2001) also did not envisage Fickian movement when they proposed their inverse Gaussian mixture model.

As *p* moves toward 0.5, implying frequent and deep particle sojourns into the subsurface, the inverse Gaussian distribution shifts toward heavy-tailed forms with small values of the shape parameter  $\phi$ . The special case of *p* = 0.5 is the drift-free situation and can be obtained as the limit distribution h(t), which arises by holding the inverse Gaussian mode constant at some value  $\xi$  and having  $\phi \rightarrow 0$ . This gives the 1-parameter transit time distribution:

$$h(t) = \left(\frac{3\xi}{2\pi t^3}\right)^{1/2} \exp\left(-\frac{3\xi}{2t}\right)$$
(3)

with cumulative distribution function:

$$H(t) = 2\Phi\left(-\left(\frac{3\xi}{t}\right)^{1/2}\right)$$
(4)

where  $\Phi$  is the standard normal integral.

Figure 2 gives a sense of the inverse Gaussian distribution shifting its shape progressively toward the heavy-tailed h(t) form as  $\phi$  decreases.

Figure 2: long-tailing behaviour of the inverse Gaussian distribution as a function of the shape parameter  $\phi$ . The vertical axis gives the value of the distributon 0.75 quantile, expressed as multiples of the modal value. The value for  $\phi = 0$  is obtained from Eq. (4).

Kingman in his comment contribution following Folks and Chhikara (1978) speculates as to possible practical application of h(t). It would be interesting if after this long period of time it found some use in hydrological transit time studies, though its application might be restricted by not having defined moments. For a single point pulse of tracer its presence would be suggested by the log of tracer concentration for large t declining linearly as a function of the log of time, with gradient -3/2.

All the above comments associated with Eqs (1) - (4) are additional to the content of the original submitted paper. My apologies for introducing new material at this late stage in the discussion period, which the Editor may wish to extend a little to allow time for some further referee input.

**I would consider revising the title**

This point was also raised by Referee #1 and Referee #2. However, I wish to keep the title (if the paper is accepted) because it does describe exactly the subject matter of the paper – which will be immediately apparent in any case from the abstract. As the Referee notes, the term "L-shaped" is not unknown in the literature, along with other various letters of the alphabet for different distribution forms.

P2, L12: Yes, but at f(0), the probability could also simply be low (i.e., not zero) particularly if there are some parcels of water with almost instantaneous exit like those falling on the recorder.

This comment is also reflected the Referee #2 comment that it is correct in the Kirchner (2001) model to include the river in the integral, and that rain may fall into the river.

Referee #3 is also happy with the concept that there can be tracer particles precisely at the measuring point at t=0 and that these particles contribute to the transit time distribution.

Referee #5 seems to concur also, with what appears to be a reference to simultaneous input and output.

And all four referees are incorrect. Rather than going through the variations on the theme one at a time, a single thought experiment is proposed which captures their arguments, following from the submitted paper. A perfect observation device records the arrival times of river-transported tracer particles which transit to the device. Of course, any tracer particles "at" the tracer device at time zero are still recorded, so the device records non-zero tracer particle frequency at exactly time zero. However, any tracer particles exactly at the observation device at exactly time zero are not part of the transit time distribution because they have not transited to the device and therefore f(0) = 0 because particles cannot move at infinite speed to the recorder. The particles which were "at" the device at time zero are simply measurement noise and no more part of the river system transit time distribution than tracer particles from a raindrop which happened to be sitting on the recorder at time zero. The discussion is analogous to considering the distribution of first passage times of a random walk from within the absorbing barrier – because then the random walk could not exist. The point here is that the hydrological sciences, and other scientific disciplines, are not free to make up their own rules for the mathematical properties of first passage time distributions.

P6, L9-10: A real problem here is that the author is ignoring a significant body of work and much of the latest work on catchment transit time where the community seems to be avoiding parametric distributions in the first place. A good example is the storage selection approach (e.g., see Rinaldo et al. 2015). There are fewer and fewer papers in the literature using parametric, stationary distributions.

Not so much ignoring it perhaps but the comment should be made in the paper along the lines that when parametric distributions are being considered etc..... and recognising that their use is declining. However, the argument in the paper makes no assumption concerning whether transit time distributions happen to be stationary or not. Also, while parametric distribution use may be declining it would seem from the comments of Referee #2 that the exponential distribution at least looks to continue as the null transit time distribution with noisy data.

Finally, I have not acknowledged one at a time the many helpful comments and corrections made by the Referee. In the event of the paper advancing further they all will of course be noted and responded to with notification to the Editor.

Folks, J.L., Chhikara, R.S., 1978. The inverse Gaussian Distribution and its statistical application--A review. Journal of the Royal Statistical Society. Series B, 40, p.263-289.

Scher, H. et al. 2002. The dynamical foundation of fractal stream chemistry: The origin of extremely long retention times. Geophysical Research Letters, 29, (5), 1061, 10.1029/2001GL014123

---

## Author Comment (AC5) · 8 Oct 2017

My thanks to Referee #5 for comment. Responses are indicated below.

The technical note focusses at a single point that in a pdf describing a breakthrough experiment in an catchment f(t=0) must be zero by necessity. Distributions fitted to breakthrough curves not fulfilling this criterion are termed "L-shaped" by the author. He proposes to avoid them.

No – distributions with f(0) > 0 are not defined to be L-shaped. See responses to Reviewer #1 and Reviewer #2, who make the same assertion. Some alternative distributions are now proposed (see response to Reviewer #4) so the "single point" aspect should no longer apply.

The argument would benefit from having its physical aspects more clearly delineated from its mathematical aspects. In many cases the problematic assumption is stationarity as a prerequisite for using pdfs for describing breakthrough curves in the first place. Furthermore, does the argument still hold when mathematical descriptions of what is termed "store" employ fractal dimensions?

The paper is essentially a mathematical argument with respect to the nature of first passage times. This argument needs no reference to stationarity. See response to Referee #4 for further comment re fractal dimensions.

It can be quite hard to show that something is impossible within a given formal framework. The part of hydrology addressed by this technical does not have such an accepted theory, tracer hydrology includes many pragmatic approaches.

It can in fact be very easy to do just that within a formal framework – see response to Reviewer #3. Pragmatism is no justification for incorrectness when there are alternatives available – see also response to Reviewer #4.

The most unclear point to me is the idealized catchment employed in the thought argument. The argument by the author seems to assign a physical storage to this ideal catchment with definite physical features.

The only physical requirement is that the catchment has spatial dimension (obviously). It is not idealized in any way and may or may not contain biota. What is idealized is the perfect instrumentation and the conceptual tracer, in order to make the point.

One could envision a second thought experiment in which the boundary of a natural catchment containing biota is entirely defined by observations of fluxes across them. Its external relations are described in classes of behavioral equivalence (similar to theoretical descriptions of computer programs) and any assignment of local internal properties becomes an over-interpretation of the available data.

I am unsure of the intended meaning here, but the thought experiment does not involve any assignment of local internal properties.

And of course, as already brought up in the ongoing discussion, input fluxes may hit the boundary producing output.

For any storage with spatial dimension an input flux can only produce output through a boundary at the same time if the particles contained within the flux travel at infinite speed.

---

## Author Comment (AC6) · 10 Oct 2017

From Referee #4

*P6, L7: Is there a question of time step or bin interval? It seems to me that for very fine bin intervals, it is possible to have transit time pdf with a mode very near t = 0 and hydrologically this will not result in transport that this very different from an L shaped distribution. The other issue is that often continuous pdfs of transit time are actually implemented discretely in time such that for times near zero, they may be indistinguishable from L-shaped distributions like a gamma with alpha < 1.*

Yes – this is very much related to bin interval. For both fine bin intervals and also continuous pdf expressions modes may be located near to zero, making little difference to any modelling process, as was noted in other responses. This would apply even more so if comparing cumulative distribution functions. The argument against the use of L-shaped distributions in such situations is purely on the basis of appropriate form – summarised in the cited comment by Kingman.

From Referee #5

*specific comments: I could not find an error in (11)*

This was discussed in responses to other reviewers and was not revisited here,

---

## Author Comment (AC7) · 10 Oct 2017

My thanks for the comment. It is interesting to see models developed as alternatives to L-shaped forms in the context of the instantaneous unit hydrograph, with zero discharge at time zero.

---

## Short Comment (SC2) · 11 Oct 2017

This technical note proposes, not just that L-shaped transit time distributions cannot exist, because "the transit time of a particle must always be with reference to a store, the transit time being some finite duration of time between particle entry and exit." Elsewhere essentially the same argument is repeated in different forms, including "... the M tracer particles placed onto the recorder at t=0 never transited through any part of the catchment system and therefore have no connection to catchment transit times," and "... it is not possible to have transit times of exactly zero because any tracer particles initially present on the recorder have never entered the store concerned. That is, they did not transit to the recorder."

[Figure]

This is characterized as "a purely conceptual argument." A better characterization would be that it is a purely definitional argument. What has happened is that the author has chosen to DEFINE transit times as necessarily being non-zero, and thus transit times of exactly zero have been excluded purely by definition.

The author's responses to several previous comments, in which he says that transit times of zero are impossible because they imply infinite velocity, repeat the same definitional argument dressed up in different clothes. The author is simply asserting that by definition transit distances cannot be zero (otherwise no "transit" has occurred), and therefore this must require some finite time.

One can easily see that this is a question of definitions rather than physical reality. For example, rain falls everywhere in a catchment, including into the stream. Assuming the detector is located in the stream, then rain can fall directly on the detector and its transit time will be zero. There is no logical or physical reason why a tracer cannot enter at the same location as the detector, and thus have a travel time of zero. Why should we assume that rain can fall everywhere in the catchment, EXCEPT at the observation point? The author's argument is simply that we should not count this as part of the transit time distribution because no "transit" has occurred.

It is important to recognize that because this is a purely definitional matter, it has exactly zero implications for the physics of water movement in the environment. In the example outlined above, for example, whether we choose to include the particles with zero transit time as part of the transit-time distribution, or not, will have exactly zero implications for how each raindrop travels to the detector or the time it takes to do so.

The physical irrelevance of the definitional inclusion or exclusion of t=0 transit times is mirrored also in its mathematical irrelevance. Whether p(t) is greater than zero for t in the range (0...x) or in the range [0...x] – that is, whether the lower bound at zero is open or closed – makes precisely zero difference to any calculations performed over any continuous interval of time, for the reason that the total probability (not probability

density, but probability) associated with the value of t=0 (or any other exact real-number value) is exactly zero. Therefore, even as a matter of rigorous mathematics, the definition that is adopted in this manuscript (that transit times can be any positive real number but not exactly zero) has exactly zero consequences.

The only way that this manuscript's argument can be consequential is if one excludes not only values of exactly t=0, but also values in some meaningfully large interval above zero. But the fundamental problem is that even if the author's argument (not really an argument, but just an arbitrary definition) were accepted, it does not establish any logical reason to reject any transit time greater than zero, no matter how small. Unless one can also rule out some finite range of non-zero transit times, the argument has no physical (or even mathematical) effect.

However, ruling out a finite range of non-zero transit times would require the manuscript to abandon its purely definitional approach to the problem, and to state that not only are transit times of zero impossible, but transit times of up to 1 minute (or 1 second, or 1 hour, or 1 something) are impossible. Doing that, however, entails a burden of proof that apparently cannot be met (because if lags of, say, 1 hour are possible, why not lags of one half hour, or one minute, or one second, or one nanosecond, or any other span of time)?

Much of the apparent force of the manuscript's definitional argument comes from a confusion between probabilities and probability densities. For example, the manuscript proposes a thought experiment where we distribute N particles over the catchment and place M particles exactly on the detector. The problem is that this creates a discontinuous and grossly nonphysical probability distribution, in which the density of particles at t=0 is infinitely higher than the density everywhere else, even infinitesimally close to t=0. As far as I know, nobody has proposed that transit time distributions might include Dirac delta functions at t=0. Thus, the context of the transit time literature, this is nothing more than a straw man argument.

In its treatment of my own work, the manuscript makes an analogous error: "An equivalent argument can be made by noting the two-parameter inverse Gaussian form of Eq. (8) of Kirchner at al. (2001). Therefore h(t) corresponds to an infinite mixture of inverse Gaussian transit time distributions. This infinite mixture distribution can be represented to any degree of accuracy as a finite mixture distribution – in this case a finite mixture of a sequence of inverse Gaussian distributions with progressively decreasing mean and variance as the tracer input point x* decreases by increments toward the observation point at x = 0."

But this is of course the central problem: saying that an infinite mixture of inverse Gaussians can be represented "to any degree of accuracy as a finite mixture distribution" is false for the argument that the author wants to make, because that argument concerns an interval that is not finite (but instead, the infinitesimal range around the exact value of zero). For this specific problem, any finite mixture distribution is an (infinitely) horrible approximation to the true distribution in the infinitesimal range around t=0. The rest of the manuscript's argument concerning my work also fails for analogous reasons.

One needs to be very careful in jumping between continuous and discrete distributions, or continuous and discrete mathematics more generally. The required degree of care has not been taken here.

The manuscript's abhorrence of transit times of zero leads to the very strange result that, for example, a gamma distribution with a shape factor of 1.000000000000001 would be declared to be realistic (because p=0 at t=0), while a gamma distribution with a shape factor of 1.000000000000000 (an exponential distribution) would be considered to be problematic because it has a finite probability density at t=0, and a gamma distribution with a shape factor of 0.999999999999999 would be regarded as anathema because its probability density climbs toward infinity as t approaches zero. This makes no sense, for two reasons. First, gamma distributions with shape factors of 1, 1+epsilon, and 1-epsilon are indistinguishable "to any degree of accuracy" (to borrow the manuscript's phrase). And second, in all three cases (and for all other gamma

distributions), the probability of t=zero is, in each case, exactly zero (as it is with any continuous probability distribution).

That last statement re-casts the point made above, namely that the definitional distinction advanced in this manuscript has no physical or mathematical consequences. If the author wants to claim that the probability of t=0 must be zero, that's fine, since the probability of ANY exact value is ALWAYS zero for ANY continuous distribution over real numbers. But then the author's argument has no physical or mathematical consequences (and any claim of one distribution being more "realistic" than another is logically empty). For the author's argument to have any consequences, he would need to exclude some finite interval of time (corresponding to some finite probability), but then he would need to justify why a non-zero transit time is impossible. In the world of continuous space and time, that would be a difficult case to make.

The foregoing arguments are completely separate from the EMPIRICAL question of whether real-world transit time distributions have peaks at lag times greater than zero. This is certainly possible (and indeed in Kirchner et al. 2001, I described cases that would generate this result). But whether this is actually true in the real world is an empirical question, to be answered, within the limits of our ability, with data.

Several research groups (including mine) are doing the hard work of making tracer measurements over very small sampling intervals in real-world catchments, and they may (or may not) find that there is a measurable non-monotonic behavior in the transit-time distribution for very short lags. But unless and until they do, it would seem wise to refrain from claiming, on the basis of arbitrary definitional premises, and in contradiction to a large body of empirical evidence, that L-shaped distributions are fundamentally impossible.

Kirchner, J.W., X.H. Feng, and C. Neal, Catchment-scale advection and dispersion as a mechanism for fractal scaling in stream tracer concentrations, Journal of Hydrology, 254, 81-100, 2001.

---

## Author Comment (AC8) · 11 Oct 2017

My thanks James for taking the time to respond. You have nicely summarised how the case for L-shaped transit time distributions could be argued. My apologies for a little delay in my reply.

---

## Author Comment (AC9) · 14 Oct 2017

My thanks to James Kirchner (JK) for his comments - responses follow below.

By way of initial comment, we can consider the JK statement:

There is no logical or physical reason why a tracer cannot enter at the same location as the detector, and thus have a travel time of zero.

It is clear, therefore that JK has no issues concerning a transit time probability density function f(t) to be such that f(0)>0, thereby allowing the possibility of L-shaped distributions. Presumably this view is also held by some number of his associates well.

It is interesting to contrast this with the comments of Referee # 1 who, though feeling that the f(0)>0 issue is of no great importance in a practical sense, makes the point that:

...we are all well aware that the property at t=0 is incorrect...

As it happens, the "all" in this case is obviously too encompassing because it evidently does not include the JK group. This is not to imply that the nature of f(0) is a topic of current debate, but there is at least a need for some degree of comment to clarify matters.

Shifting now to the hydrological argument, which might be termed the raindrop-on-recorder situation, summarised by JK as:

Assuming the detector is located in the stream, then rain can fall directly on the detector and its transit time will be zero.

There is no question that the rain can fall directly onto the recorder. Indeed, a conceptual raindrop could already be present on the recorder even before t=0. However, all those tracer particles (one or more) present at the recorder at t=0 cannot be part of a transit time distribution with zero transit times because there is no transit involved. Assume for now that JK is in fact correct here. Transit time distributions would then be comprised of the transit times of two different populations of tracer particles: (i) particles which have transited, and (ii) particles which have not transited. The transit time distribution would then consist of (i) a probability density function which does not integrate to 1, and (ii) a fixed probability (not probability density) at t=0. With respect to such a distributional monstrosity, I can think of no better description than that given by JK:

The problem is that this creates a discontinuous and grossly nonphysical probability distribution, in which the density of particles at t=0 is infinitely higher than the density everywhere else, even infinitesimally close to t=0.

I should add, incidentally, that the thought experiment at the start of the submitted paper was purely in anticipation (correctly, as it turns out) of the raindrop-on-recorder argument being raised to make a case that L-shaped transit time distributions are conceptually possible. With that particular argument hopefully now disposed of, it would not be reproduced in any final paper.

At this point we can presumably agree that the discussion centres on definitions and probability density functions, and not on probabilities. In regard to definitions, I seem to stand accused by JK of inserting some sort of personal definition to exclude zero transit times, and then using that definition for my own ends to consequentially define the impossibility of L-shaped transit time distributions.

To demonstrate my lack of personal input here, we need to first recall just what is being transiting "to" in transit time distributions. As far as I am aware, hydrological transit time distributions always have transits terminating as a death process. For example, a tracer particle on reaching a recorder does not pop back out of the recorder, perhaps to be recorded a second time. That is, hydrological transits terminate when an absorbing barrier is reached.

Various forms of transits to absorbing barriers are much-researched in stochastic analysis, as HESS readers will be aware. However, for the purposes of the present discussion it is sufficient to refer briefly just to the discrete random walk between a reflecting and an absorbing barrier (Weesakul, 1961), as mentioned in the submitted paper.

Specifically, given an absorbing barrier at x = 0 and a reflecting barrier at x = b, a particle starts its transit at some point *u*, where  $0 < u \le b$ . Once motion is initiated, the particle moves about by discrete random motion until the transit is terminated when reaching the absorbing barrier for the first time.

The important factor here is the definition of the range of starting points for a transit. Was Weesakul making an arbitrary decision for his own ends when he defined  $0 < u \le b$  and as opposed to  $0 \le u \le b$ ? Of course not. Our mathematical colleagues are meticulous in their definitions and the specification of  $0 < u \le b$  arises from the basic logic that transits to an absorbing barrier cannot be initiated from within the absorbing barrier. This property is independent of any supposed "definition" of mine and will of course still remain regardless of whether the submitted paper is accepted or not.

It follows that because u > 0 when t = 0, transit time distributions to absorbing barriers must have f(0) = 0 because there are no zero-time transits. And it follows in turn that derivations in hydrology which supposedly obtain L-shaped transit time distributions must contain an error at some point in their argument, which will have influence on the form of the derived distributions near t = 0. This is not a mathematical error in algebra, but an error in the specification of the problem to be solved.

There is nothing to stop *u* from being located arbitrarily close to 0 but the requirement of u > 0 always remains. This relates as well to the inverse Gaussian mixture discussion for the L-shaped transit time distribution of Kirschner et al (2001). Regardless of whether we have a finite or infinite mixture of inverse Gaussian distributions, each inverse Gaussian distribution has the property of f(0) = 0. The summation of even an infinite number of zeroes at t = 0 still yields a mixture transit time probability density of zero at t = 0.

The conclusion (and title) of the paper remains exactly unchanged – transit time distributions are not L shaped. And they are not L-shaped because they are fundamentally impossible. As responsible editors of an international journal, the HESS editors will never consider publishing a paper whose very title they see to be nothing more than an unproven assertion. The publication decision is therefore of interest.

I would like to note that the paper was not written for the purposes of being critical of any previous publication. The paper stands on the basis of its own argument. Some critical reference to Kirchner et al (2001) was simply unavoidable because that paper is arguably the best-known published hydrological work deriving an L-shaped distribution. If no reference to it had been made then the referees would most probably have advocated rejection, pointing out that Kirchner et al (2001) had demonstrated via analytical argument that L-shaped transit time distributions can exist. If the HESS publication process advances then my personal preference would be for having a shorter paper avoiding listing specific criticisms of other works and focus on the positive aspects – while fully appreciating that JK sees zero of the latter.

Turning to one further specific point, with respect to the comment:

.. the definitional distinction advanced in this manuscript has no physical or mathematical consequences

I can only fully agree – at least to the extent that the paper makes no pretence to provide new insights into the physics of moving water in the environment and certainly does not offer anything of mathematical consequence. However, what is does demonstrate (as is well-known according to Referee #1) is that L-shaped transit time distributions cannot exist as mathematical entities, so L-shaped parametric transit time distributions would seem inappropriate for application to transit time data.

The remaining JK comments relate to aspects of observational limitations. These were discussed as part of the response to Referee #4 and are not repeated here.

It is fully appreciated that this response will be totally unsatisfying to JK but I would like to offer my thanks for his contribution in any case. The submitted manuscript has now (in effect) had six referee comments so there can certainly be no issue over it not having had wide-ranging review.